# A College Fast-Food Environment and Student Food and Beverage Choices: Developing an Integrated Database to Examine Food and Beverage Purchasing Choices among College Students

**DOI:** 10.3390/nu14040900

**Published:** 2022-02-21

**Authors:** Elizabeth F. Racine, Rachel Schorno, Shafie Gholizadeh, Morium Barakat Bably, Faizeh Hatami, Casey Stephens, Wlodek Zadrozny, Lisa Schulkind, Rajib Paul

**Affiliations:** 1Texas A&M AgriLife Research, Texas A&M University, El Paso, TX 79927, USA; 2Department of Public Policy, University of North Carolina at Charlotte, Charlotte, NC 28223, USA; rachelschorno@gmail.com; 3Department of Computer Science, Computing and Informatics, University of North Carolina at Charlotte, Charlotte, NC 28223, USA; shervin.gholiza@gmail.com; 4Department of Public Health Sciences, University of North Carolina at Charlotte, Charlotte, NC 28223, USA; mbably@uncc.edu (M.B.B.); csteph29@uncc.edu (C.S.); rpaul9@uncc.edu (R.P.); 5Department of Geography and Earth Sciences, University of North Carolina at Charlotte, Charlotte, NC 28223, USA; fhatami@uncc.edu; 6Department of Computer Science, University of North Carolina at Charlotte, Charlotte, NC 28223, USA; wzadrozn@uncc.edu; 7Department of Economics, University of North Carolina at Charlotte, Charlotte, NC 28223, USA; lschulki@uncc.edu

**Keywords:** emerging adults, food sales data, integrated dataset, healthy food score, university food environment, fast-food restaurants

## Abstract

Universities typically offer residential students a variety of fast-food dining options as part of the student meal plan. When residential students make fast-food purchases on campus there is a digital record of the transaction which can be used to study food purchasing behavior. This study examines the association between student demographic, economic, and behavioral factors and the healthfulness of student fast-food purchases. The 3781 fast-food items sold at the University of North Carolina at Charlotte from fall 2016 to spring 2019 were given a Fast-Food Health Score. Each student participating in the university meal plan was given a Student Average Fast-Food Health Score; calculated by averaging the Fast-Food Health Scores associated with each food and beverage item the student purchased at a fast-food vendor, concession stand, or convenience store over a semester. This analysis included 14,367 students who generated 1,593,235 transactions valued at $10,757,110. Multivariate analyses were used to examine demographic, economic, and behavioral factors associated with Student Average Fast-Food Health Scores. Being of a low income, spending more money on fast-food items, and having a lower GPA were associated with lower Student Average Fast-Food Health Scores. Future research utilizing institutional food transaction data to study healthy food choices is warranted.

## 1. Introduction

College students living on campus are a unique population—transitioning from home to a relatively independent environment. As children age, their eating habits tend to become less healthy and their preferences change, often leading to a greater intake of fast food [1]. University students report poor dietary intake [2,3,4,5], and it is well-documented that transitioning to college is associated with excess weight gain [6,7,8]. The college food environment allows students to determine their own food choices for possibly the first time in their lives as they choose where, when, and what to eat.

In the past 5 years, a number of studies have been published examining the dietary behaviors of college students. Many are conducted in Australia [9,10,11,12,13,14], New Zealand [15,16], and Europe [2,17,18]. Poor dietary behaviors among college students are associated with lower academic achievement [9,19], poorer resilience [10], higher psychological distress [10], being male [2,3,11], being younger [12,20], being less physically active [6,17], and having lower socioeconomic status [20,21].

When college students make food choices, they are often selecting from foods offered on-campus. In the US there are no regulations for the healthfulness of university food environments as there are for public primary and secondary schools [22]. There are guidelines that universities can choose to follow from organizations such as the Partnership for America [23] Healthy Campus Initiative and the Menus of Change University Research Collaborative.

A few studies examining college student eating behavior found students that purchased food on-campus more frequently had poorer diet quality [13,16,20]. Additional research suggests that much of the food sold on-campus is not healthy [15]. Students report that there is a lack of tasty, healthy, affordable foods available [14,15]. Yet, it is not clear whether offering healthy food items will ensure selection of those items. A study by Lachat in 2009 assessed the foods purchased in a university dining hall by taking a picture of the student’s food tray once they made their selections from a cafeteria-style food line [3]. The authors compared the healthfulness of the foods offered at the dining hall to the foods purchased and found that the students purchased the less healthy items on the cafeteria line more frequently than they purchased the healthier items [3].

College food environment and student food choice research rarely uses food sales data to assess student food choice; except in the case of vending machine research [11]. Much of the student food research to date relies on student reports via a variety of methods such as surveys [11,13,15,17,20], 24 h recall [2,18], or direct observation [3].

Many universities and colleges in the United States hire private food service contractors, while others manage their own food service operations. Most colleges offer multiple options for on-campus dining, and the meal plans for students often include both dining hall access and some form of a declining balance funds system. These options allow students to choose between eating their meals at the dining hall and purchasing meals or individual food and beverage items at on-campus fast-food restaurants. At the University of North Carolina at Charlotte, a southeastern urban university, there are over 20 fast-food restaurants, concession stands, and convenience stores on campus, offering over 3700 food and beverage items. There are a variety of fast-food options, including coffee shops such as Starbucks and Peet’s, traditional fast-food restaurants such as Wendy’s and Chick-fil-A, and market-style convenience stores where students can purchase prepackaged sandwiches, snacks, and cooked food to go as well as concession stands that operate during sporting and other campus events. There are also two campus dining halls with all-you-can-eat buffet-style meals. Residential students can use their meal plan to purchase meals at the dining halls or to purchase food and beverage items at fast-food restaurants, concession stands, or convenience stores on campus.

In the United States about 40% of residents between the ages of 18–24 years attend a postsecondary education program; that equates to approximately 15 million college students [24]. As these students learn to live independently and develop healthy behaviors it is important to examine the role that the college food environment plays in their nutritional development. The purpose of this manuscript study is to determine which demographic, economic, and behavioral factors are associated with the healthfulness of fast-food choices among students attending a large, urban university in the southeastern United States.

## 2. Materials and Methods

### 2.1. Dataset Development

UNC Charlotte Integrated Food Sales Dataset was developed in 2016 by an interdisciplinary team of researchers in the areas of public health, computer science, public policy, and economics. The university division of Auxiliary Services maintains an electronic record of the food and beverage transactions that occur on campus. Students participating in the university meal plan use their student identification (ID) card to purchase food on campus. A student meal plan consists of a certain number of “meal swipes” per semester and a certain amount of “declining balance dollars” (hereafter DBD) per semester. The university offers a few meal plans; each provides a certain number of meal swipes and DBD. A meal swipe is used for a meal at a cafeteria style dining hall that offers the student a wide variety of food and beverage choices and is all-you-can-eat style. DBD are funds that can be used at the fast-food restaurants, concession stands, and convenience stores on campus. This analysis focuses specifically on the purchases made by students using their DBD at the campus fast-food retailers, concession stands, and convenience stores (hereafter referred to as fast food).

Each food or beverage electronic transaction at a university fast-food outlet captures the student’s ID number. This ID number is the same number used to identify the student for a variety of university purposes. The research team worked with the university’s department of Auxiliary Services to acquire the food and beverage transaction data retroactively to fall 2013 and continues to collect transaction data at the end of each academic year.

The food and beverage transaction data include factors regarding student purchases at university-based fast-food restaurants made with their declining balance dollars, such as date and time of transaction, price of item, balance of declining balance account, name of item, and modifications to the item (e.g., no lettuce, extra cheese, etc.).

Once provided with food and beverage transaction data, the research team worked with other university departments to acquire more details about the meal plan students’ demographics, grade point average (GPA), residential environment, income status, and visits to a recreational facility. Additionally, the research team obtained some nutrition information for the food and beverage items (*n* = 3781). The US Food and Drug Administration requires all restaurants to have the following nutrition information available to customers, hereafter referred to as FDA Restaurant Nutrients: total calories, calories from fat, total fat in grams, saturated fat in grams, trans fat in milligrams, cholesterol in milligrams, sodium in milligrams, total carbohydrates in grams, fiber in grams, sugars in grams, protein in grams. The nutrition information for food and beverage items was acquired from the campus food service registered dietitian, as well as from the campus retailer official websites. A separate dataset was built containing all the food and beverage items available at the fast-food restaurants, concession stands, and convenience stores on campus during the time the sales data were collected and linked the nutrient information to each item.

To estimate the healthfulness of the food and beverage items, the research team used the nutrition information described above to construct the Fast-Food Health Score. The Fast-Food Health Score applies dietary recommendations from the 2020–2025 US Dietary Guidelines for Americans for total fat, saturated fat, total carbohydrates, fiber, protein; the 2005 National Institute of Medicine Dietary Reference Intakes for Water, Potassium, Sodium, Chloride, and Sulfate for sodium, and the World Health Organization Guideline: sugar guidelines, Table 1 [24,25]. Virtually no food and beverage items included trans fat, therefore the Fast-Food Health Score does not include a component for trans fat [24]. The nutrients in the food or beverage item are evaluated in relation to the item’s calories. Each food and beverage item were evaluated on a seven-point scale. A food or beverage received one point for each of the seven FDA Restaurant Nutrient attributes classified as healthy. A nutritional component was classified as healthy if the amount of that nutrient in the food or beverage fell within recommended standards for a healthy diet, as shown in Table 1. The greater the number of points (ranging 0–7) on the Fast-Food Health Score (FFHS) scale the healthier the food or beverage item.

UNC Charlotte Integrated Food Sales Dataset includes 16 semesters of data (fall 2013–spring 2021). However, the data presented here includes six semesters (fall 2016, spring 2017, fall 2017, spring 2018, fall 2018, and spring 2019); these are the semesters that include information on Bojangles purchases, a popular new fast-food restaurant on campus, information on the recreational facility use visits, and complete dietary score measures. Data from fall 2019 to spring 2021 are not included for a few reasons (1) data cleaning is not complete for these semesters, and (2) university food sales operations changed temporarily because of the COVID-19 pandemic. Specifically, the number of students living on-campus decreased by two-thirds, many of the fast-food restaurants closed, and the ordering process moved from face-to-face to online only.

### 2.2. Setting/Participants

UNC Charlotte is a large urban university in the southeastern US. The cost of attendance is about $24,000 per year, similar to other public universities in the US [26]. UNC Charlotte serves a diverse student population by income level, first generation attending college status and race/ethnicity [27]. Residential and some commuter students at the university purchase meal plans. Freshman students who live on campus are required to purchase a meal plan, as are upper-class students living in residential halls that do not contain a kitchen area. These students are allowed to choose from meal plans that include varying quantities of meal swipes and DBD. Both the card swipes on the purchased meal plans and the DBD expire at the end of each semester, and do not carry over for winter or summer breaks. Most students in the study data are between the ages of 17 and 22.

Approximately 5500 students are participating in the meal plan per semester; representing approximately 20% of all students enrolled at the university. The study population consists of all students attending the university who purchased a meal plan for at least one semester during the academic years 2016 through to 2019. Many students participate in the meal plan for multiple semesters. If a student is enrolled in the meal plan for three semesters, information for that student (GPA, Student Average Fast Food Score, residence hall, etc.) is recorded as a unique observation for each semester. In the current study, 35,449 total student observations represent 14,367 unique students.

### 2.3. Measurement

The outcome variable in this study is the Student Average Fast-Food Health Score (Student Average FFHS) that has been calculated for each student by averaging the FFHS for each of the food and beverage items purchased at fast-food venues for that semester. This score is averaged for the purchases at outlets (fast food, concession, convenience) using DBD only; it does not include the food consumed at dining halls, as there are no current means of obtaining the exact food and beverages a student selected when using meal plan swipes due to the buffet style dining hall environment.

Independent variables are categorized as demographic, economic, and behavioral. Demographic variables are race/ethnicity as categorized by the university (African American, Asian or Pacific islander, Hispanic, White, or any other race) [28], International student (yes/no), and age [28]. Economic variables are being low income, measured by Federal Pell Grant recipient status (yes/no) [29] and money (measured in dollars) spent on fast food in a semester (continuous) [30]. Behavioral variables are semester grade point average (0–4) [31,32], frequency of dining hall use over a semester (continuous) [33], and average frequency of visits to a recreational center over a semester (continuous) [34].

### 2.4. Data Analysis

An FFHS was calculated for the 3781 food and beverage items purchased throughout the fall 2016-spring 2019 academic years. The average and median scores were also calculated. The Student Average FFHS was calculated for 35,449 student observations.

Descriptive statistics are presented for the complete population of students and stratified by sex. To address the primary research question, to determine which demographic, economic, and behavioral factors are associated with the healthfulness of fast-food choices among students participating in the meal plan, we used multiple linear regression models with heteroscedastic variances inversely proportional to the number of food items purchased by each student. The models included the Student Average FFHS as the dependent variable, and the independent variables listed above. All analyses used a significance level of *p* < 0.05, and all data were analyzed using R.

## 3. Results

The distribution of Student Average FFH Scores for the 35,449 student observations is presented in Figure 1. The distribution of the 3781 food and beverage items purchased throughout the fall 2016-spring 2019 academic years is presented in Figure 2. Most food and beverage items had a FFHS of 2 or 3. Here are examples of food and beverage items by score: Monster Energy Drink 16 oz had a score of 0, Bojangles Sausage Biscuit had a score of 1, Chik-fil-A Fried Chicken Delux had a score of 2, Peete’s Bagel and Cream Cheese had a score of 3, Wendy’s Smoky Honey Mustard Flatbread Chicken Sandwich had a score of 4, Subway Buffalo Chicken 12” sandwich had a score of 5, and the Peet’s Turkey Sandwich on Wheat Bread had a score of 6.

Between fall 2016 and spring 2019, meal plan students spent $10,757,110 with DBD generated from 1,593,235 transactions. Table 2 shows the characteristics of the students in the study. Students spent on average $303 per semester with DBD. Most students self-identified as White (59%) or African American (22%). Half of the students (54%) had a GPA of 3.01–4.0. Finally, 33% of students on the meal plan were low income (Table 2). Bivariate analyses confirm that the mean Student Average FFH score for females is slightly higher than the mean Student’s Average FFH score for males (2.87 compared to 2.82, with *p* ≤ 0.0001), Figure 2.

In adjusted analyses (Table 3), females and males with a higher GPA and visiting the dining hall more frequently were associated with having a higher Student Average FFHS. Spending more money on fast food and being low income were associated with a lower Student Average FFHS. Compared to White students, African American and Other race/ethnicity students had a lower Student Average FFHS.

Differences by sex were found in the areas of age, recreational facility visits, international student status, and race/ethnicity. Visiting the recreational facilities more frequently was associated with a lower Student Average FFHS among females yet a higher Student Average FFHS among males. Being older was associated with a lower Student Average FFHS for females and a higher Score for males. Asian and Hispanic males had a higher Student Average FFHS compared to White males. Among female students, no difference in Student Average FFHS was found between White, Asian/Pacific Islander, and Hispanic students. Male international students had lower Student Average FFHS compared to noninternational male students. No difference in Student Average FFHS was found between international and non-international female students.

## 4. Discussion

This study describes the development of the UNC Charlotte Integrated Food Sales Dataset. A Fast-Food Health Score was generated for the food and beverages sold. Then, the Fast-Food Health Scores for all the items purchased over a semester were averaged for each student, resulting in a Student Average Fast-Food Health Score. Multivariate analyses examined the demographic, economic, and behavioral factors associated with Student Average Fast-Food Health Score.

To our knowledge, UNC Charlotte is the first university in the US to utilize student food sales data to better understand the food purchasing behaviors of college students. The majority of research examining college student food purchasing behavior used student self-reported data [12,14,15,17,20]. This study provides an alternative source of dietary choice information that can be used to assess student preferences, assess the impact of food environment or policy changes on food purchasing behavior, or evaluate the impact of nutrition education interventions.

An algorithm was constructed utilizing nutrition information that the FDA requires restaurants to have available to customers to construct the Fast-Food Health Score. Further research to examine the utility of the FFHS among public health nutrition researchers and practitioners is warranted.

Similar to previous research, we found that the healthfulness of many food items sold on campus was poor [3], and the healthfulness of most food items purchased was also poor [3]. The majority of food and beverage items fall within the Fast-Food Health Score of 2–3. This parallels closely with the Student Average Fast-Food Health Score of approximately 2.8. If the number of fast-food items with higher Fast-Food Health Scores increased, would the Student Average Fast-Food Health Score also increase? Further research to study this would be useful.

As the students aged, the Student Average Fast-Food Health Scores decreased for females yet increased for males. Previous research found that older students had better diet quality measures yet did not stratify their analyses by sex [11,20]. It would be useful to follow young adults’ diet quality more closely as they develop from their late teens through to mid-20s, to better understand the relationship between development and dietary choice. Additionally, among both males and females, the more funds used to purchase fast-food items, the lower Student Average Fast-Food Health Scores. This finding is consistent with previous research [12,14,16,17]. Further research to examine the relationship between frequency of food intake and diet quality may be warranted. It may be that students that utilize on-campus food sources more frequently have less food preparation knowledge, or poorer time management or financial management skills. However, it is interesting that the students that used the dining hall facilities more frequently had higher Student Average FFHS. These students may be more health conscious and therefore prefer the buffet style foods in the dining halls which may have more healthy options. Then, when they do use their DBD, they select relatively healthy foods compared to other students.

Consistent with previous research [2,3,12,17], our study found that males had a lower Student Average Fast-Food Health Score compared to females. However, while the difference was statistically significant the scores were low for both males and females. Our study found that a greater academic achievement (higher GPA) was associated with higher Student Average FFHS among both male and female students. This finding is consistent with previous research [9,19]. Are students that are stronger academically better educated about nutrition and more attentive to food choice? Or does healthier dietary intake help students academically? More research is needed to understand this relationship.

A few factors influencing Student Average FFS differed by sex. For instance, more visits to the recreational center were associated with lower Student Average Fast-Food Health Score among females yet higher Scores among males. To our knowledge, the relationship between physical activity frequency and diet quality among university students by sex has not been previously reported. However, research by Sprake and colleagues (2018) found that higher levels of physical activity were associated with better diet quality. Additionally, the finding that international students that were male had lower Student Average FFHS compared to non-international male students is interesting. How different is the US college food environment compared to other countries? How do international students react to a typical US college food environment? Further research could help us understand the extent to which an individual’s dietary choices change when their environment changes. For example, when transferring from student status in one country to another.

There were limitations to this study. The study may not be generalizable to students attending other colleges or universities, particularly those in other countries. Students purchasing a meal plan at UNC Charlotte may be different on both observables and unobservable facets compared to UNC Charlotte students not purchasing a meal plan. Students may make healthier food choices in the dining hall venues. We did not have information regarding the food selected by students in the main dining halls on campus. Additionally, the dataset may not capture all fast-food purchases; particularly purchases made with alternate forms of payment, such as cash or credit card or purchases made at restaurants off campus. The nutrition information used to generate the Fast-Food Health Score was limited to seven nutrients, those required by the FDA, so future research linking fast-food items to more nutrients may be useful.

This study had several strengths. The study uses a novel data source. The analysis includes all the students that participated in the university meal plan. Our study also had a relatively equal balance of male and female students and includes a diverse population of students by income and race/ethnicity. We used food sales data to estimate diet quality as opposed to student self-reporting. Our study utilized sales data captured from over 20 fast-food restaurants

## 5. Conclusions

While there were factors that were either positively or negatively associated with Student Average FFHS, the differences attributable to these factors were small. Highlighting the finding that most of students’ scores were quite low regardless of student characteristics. Public health nutrition faculty based at colleges and universities are encouraged to work with university administration to assess the campus food environment, suggest modifications, then measure the impact of those modifications on food purchasing behavior and student health and well-being.

Colleges serve a large number of people, including students as well as faculty and staff. As more and more university operation systems leave an electronic record, there are a number of food environment research opportunities available. Other universities are encouraged to develop databases similar to the UNC Charlotte Integrated Food Sales database to study the impact of environmental, policy, operations, and demographic changes on dietary choice.

## Figures and Tables

**Figure 1 nutrients-14-00900-f001:**
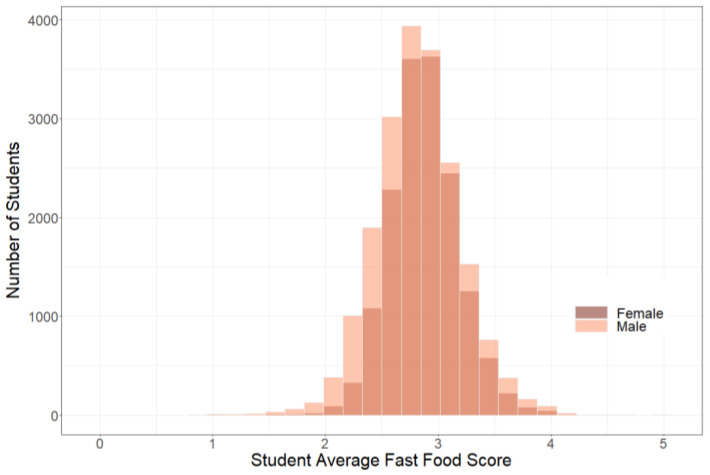
Student Average Fast-food Health Food and Beverage Score by Sex, N = 35,449.

**Figure 2 nutrients-14-00900-f002:**
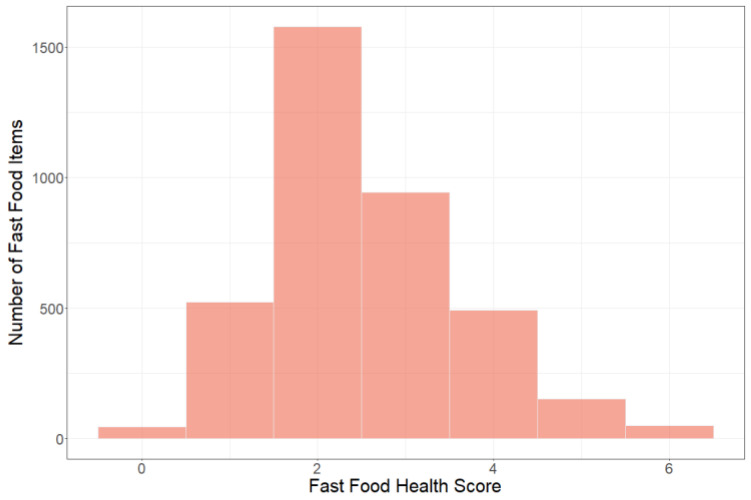
Distribution of food and beverage items (N = 3781) by Fast-Food Health Score.

**Table 1 nutrients-14-00900-t001:** Fast-Food Health Score (FFHS) Algorithm.

Fast-Food Health Score (FFHS) Components ^1^	Health Point	Affected Menu Items
Total fat is between 20% and 35% of calories	1	17.1%
Saturated fat less than 10% of calories	1	52.2%
Sodium less than 1.15 mg for every calorie	1	53.8%
Total Carbs between 45% and 65% of calories	1	30.4%
1.4 g or more fiber for every 100 calories	1	28.1%
Sugars less than 10% of calories	1	36.8%
Protein is 10–30% of calories	1	33.8%
Total FFHS Range	0–7	

^1^ Number of Food and Beverage Items scored = 3781.

**Table 2 nutrients-14-00900-t002:** Student Average Fast-Food Healthy Food and Beverage Score (FFHS) and Student Demographic, Economic, and Behavioral Characteristics 2016–2019, Stratified by Sex N = 35,449.

Variable	Total Student Observations *N* = 35,449	Female *n* = 15,730 (44.4%)	Male *n* = 19,719 (55.6%)
Student Fast-food Healthy Food Score Mean (Standard Deviation)	2.85 (0.35)	2.87 (0.31)	2.82 (0.38)
Age Median (Interquartile Range)	19 (2)	19 (2)	19 (2)
Race/Ethnicity			
African American Population (Column %)	7476 (22)	4130 (27)	3346 (17)
Asian Population (Column %)	1798 (5)	719 (5)	1079 (6)
Hispanic Population (Column %)	2558 (7)	1266 (8)	1292 (7)
White Population (Column %)	20,457 (59)	8052 (52)	12,405 (64)
Other Population (Column %)	2581 (7)	1313 (8)	1268 (6)
International			
Yes Count (Column %)	579 (2)	250 (2)	329 (2)
No Count (Column %)	34,870 (98)	15,480 (98)	19,390 (98)
Low Income Population (Total Population %)	11,753 (33.1)	5943 (37.8)	5810 (29.5)
Money Spent on Fast Food (Declining Balance Dollars) per Semester (Hundreds of Dollars) (Column %)	303.45 (208.28)	315.73 (206.12)	293.65 (209.46)
Semester GPA			
0.01–2.0 GPA Population (Column %)	3535 (13)	1102 (9)	2433 (15)
2.01–3.0 GPA Population (Column %)	9198 (33)	3421 (29)	5777 (37)
3.01–4.0 GPA Population (Column %)	15,082 (54)	7509 (62)	7573 (48)
Number of Dining Hall Visits per Semester Mean (Standard Deviation)	44.94 (27.93)	48.24 (27.69)	42.3 (27.86)
Number of Recreation Center Visits Per Semester Mean (Standard Deviation)	17.72 (30.7)	12.21 (22.46)	22.12 (35.35)

**Table 3 nutrients-14-00900-t003:** The Association between Demographic, Economic, and Behavioral Factors and Students’ Average Fast-Food Health Score (FFHS), N = 35,449.

Variables	Female	Male
Age	−0.0104 **	0.1297 **
Race/ Ethnicity		
African American	−0.0302 **	−0.0204 *
Asian/Pacific Islander	0.0123	0.0897 **
Hispanic	0.0149	0.0514 **
Other	−0.0204 *	−0.0356 *
International	0.0079	−0.324 **
Low Income	−0.0097 *	−0.0246 **
Money Spent on Fast Food (Declining Balance Dollars) per Semester (hundreds of dollars)	−0.0106 **	−0.0114 **
Semester GPA	0.0253 **	0.1098 **
Number of Dining Hall Visits per Semester	0.0014 **	0.0067 **
Number of Recreational Facility Visits per Semester (per 10 visits)	−0.00208 *	0.0022 *

** *p*< 0.01, * *p* < 0.05.

## Data Availability

The data supporting the conclusions of this study are available upon reasonable request and under the supervision of Wlodek Zadrozny.

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
