# Peer review of "A College Fast-Food Environment and Student Food and Beverage Choices: Developing an Integrated Database to Examine Food and Beverage Purchasing Choices among College Students"

_nutrients, 2022, doi:10.3390/nu14040900_

Round 1
Reviewer 1 Report
I want to congratulate the authors for the work they have done, which I believe can be accepted for publication once the elements that I am going to break down are taken into account:
The authors must clearly present the objective of the work. In the title, as well as in the purpose 1 of the work, relevance is given to the development of a database. If the objective of the work is that, the work should explain this development in greater detail and present the case. However, line 191 indicates that the primary research question is “to determine which demographic, economic, and behavioral factors are associated with the healthfulness of fast-food choices”. This last one seems really objective of the work.
In line 50 the authors indicate that "there is no research examining the food purchaising behaviors of college residential students." Yes, there are such studies, at the University level.
Line 150 indicates that “university food sales operations changed drastically as a result of the Covid-19 pandemic”. Assuming this statement is correct, one wonders if the work does not lose relevance by presenting results up to the year prior to the pandemic. I do not question the relevance of the work, but I invite the authors to assess the suitability of including said statement.
The paper analyzes in detail the effectiveness of a program to monitor the diet of university students. For this reason, it would be interesting to present in a section the types of existing programs and the application they have had in the university context.
The authors indicate as a limitation that only "have data for the fast-food restaurants, and these data are only available for purchases made with DBD funds on stidents ID". I believe that this element can be converted into a Strength of the work, without the section that I recommend in the previous comment focusing on such elements.
The element that can be a limitation is that the participants “are upper class students”. This element must be taken into account when evaluating the results.
Author Response
Reviewer 1
R1. Comment 1: The authors must clearly present the objective of the work. In the title, as well as in the purpose 1 of the work, relevance is given to the development of a database. If the objective of the work is that, the work should explain this development in greater detail and present the case. However, line 191 indicates that the primary research question is “to determine which demographic, economic, and behavioral factors are associated with the healthfulness of fast-food choices”. This last one seems really objective of the work.
Authors’ response: We agree with your point. In the introduction we removed objectives 1-3.
R1. Comment 2: In line 50 the authors indicate that "there is no research examining the food purchasing behaviors of college residential students." Yes, there are such studies, at the University level.
Authors’ response: Thank you for pointing this out. We updated the introduction and discussion with the inclusion of recent research. See lines 42-70 and 289-320
R1. Comment 3: Line 150 indicates that “university food sales operations changed drastically as a result of the Covid-19 pandemic”. Assuming this statement is correct, one wonders if the work does not lose relevance by presenting results up to the year prior to the pandemic. I do not question the relevance of the work, but I invite the authors to assess the suitability of including said statement.
Authors’ response: We added more detail to this statement to clarify how the food environment changes temporarily as a result of the COVD-19 pandemic. See lines 167-170.
R1. Comment 4: The paper analyzes in detail the effectiveness of a program to monitor the diet of university students. For this reason, it would be interesting to present in a section the types of existing programs and the application they have had in the university context.
Authors’ response: We’re not sure we understand the comment. We assume the reviewer is asking for a paragraph detailing the ways college student food purchase data has been collected in previous research. In response, we did add a paragraph describing the data collection methods used in previous research. See lines 71-74.
R 1. Comment 5: The authors indicate as a limitation that only "have data for the fast-food restaurants, and these data are only available for purchases made with DBD funds on students ID". I believe that this element can be converted into a Strength of the work, with the section that I recommend in the previous comment focusing on such elements.
Authors’ response: Thank you for the suggestion. Once we reviewed the literature more thoroughly, we realize that much of the research to date relys on student respondents reporting their intake. While in this study, all the food purchased by students is captured. We edited this “limitation” and highlighted it as “strength. See lines 345-346.
R1. Comment 6: The element that can be a limitation is that the participants “are upper class students”. This element must be taken into account when evaluating the results.
Authors’ response: Compared to college students in many other countries, the students at UNC Charlotte are likely upper class (higher SES). However, in comparison to other 4-year universities in the US, UNC Charlotte is considered pretty diverse with a third of the students low income, other 30% first generation college students, and over half identifying as a racial/ethnic minority. We added some clarification regarding the demographic makeup of the UNC Charlotte student population in lines 172-175 in the methods section and in lines 334-335 and 346-348 in the discussion section.
Reviewer 2 Report
Comments in the attachment.

Author Response
Reviewer 2
R2. Comment 1: In the ‘Introduction’ section, the literature review on the determinants of nutritional choices of student youth should be supplemented with current items from literature on the subject.
Author Response: Thank you for your suggestion. We added a few paragraphs to the introduction to better describe the literature. See lines 42-70
R2. Comment 2: In the ‘Results’ section, Table 2 should be corrected. Age data is given as the median and standard deviation (SD), meanwhile SD is the mean (M) and the median is the quartile deviation (QD). In addition, the sum of the percentages (%) for Race and Female GPA does not add up to 100.00%.
Author Response: Thank you for your comment. Interquartile range is added to the table instead of the standard deviation. Sum of the percentages not adding up to 100 were due to the issue of rounding numbers with high precisions. This was resolved using the largest remainder method.
R2. Comment 3: For greater clarity of Table 2, the measurement units (n,%) for the variables: Race, International, Pell Grant and GPA, should be placed next to them, because their placement in the table header suggests that these measures apply to all variables (which is not true as some are expressed in other descriptive statistics - M, Me).
Authors’ Response: Thank you. The table is revised as suggested and it is clearer now.
R2. Comment 4: In the description of Table 2, the entries should be corrected: p = 0.0000 to p <0.0001.
Authors’ Response: We made this change.
R2. Comment 5: The ‘Discussion’ section should be elaborated on so that it actually is of discussion nature (with regard to the references). In the present version, it is rather a discussion of the results.
Authors’ Response: We elaborated on the discussion as suggested.
R2. Comment 6: The ‘Conclusions’ section is missing.
Authors’ Response: We added a short conclusion section
Reviewer 3 Report
I started to review with interest the article, and I noticed from the beginning that authors have not sufficiently combed the literature, for instance (Lines 59-64), they say:
"Additionally, limited studies are focusing on the healthfulness of the foods college students are eating [10-11]. Horacek et al. (2013) discuss the on-campus dining environment at fifteen higher education institutions; they look at the healthfulness of the food items offered, but do not investigate the effects that these food items have on student dietary intake [10]. Similarly, Driskell et al. (2006) look at the fast-food eating habits of university students but also do not consider the health effects – current or lasting – of this level of fast-food consumption [11]."
The studies cited are relevant to the topic, but the authors miss more recent studies, which I conducted myself (2018 - https://doi.org/10.1371/journal.pone.0197874 and 2021- https://doi.org/10.1017/S0007114520004390) that examine the healthfulness of foods consumed by college students, as well as some economical considerations.
My main concern was not this, but as I continued to read the methods, I see that the authors are using linear regression for a score variable of max 7 points. The distribution of such variable requires careful statistical analysis. Using linear regression in this case is not statistically and epidemiologically sound and the authors should consult with a statistician.
Author Response
Reviewer 3
R3. Comment 1: I started to review with interest the article, and I noticed from the beginning that authors have not sufficiently combed the literature for instance (Lines 59-64), they say:
"Additionally, limited studies are focusing on the healthfulness of the foods college students are eating [10-11]. Horacek et al. (2013) discuss the on-campus dining environment at fifteen higher education institutions; they look at the healthfulness of the food items offered, but do not investigate the effects that these food items have on student dietary intake [10]. Similarly, Driskell et al. (2006) look at the fast-food eating habits of university students but also do not consider the health effects – current or lasting – of this level of fast-food consumption [11]."
The studies cited are relevant to the topic, but the authors miss more recent studies, which I conducted myself (2018 - https://doi.org/10.1371/journal.pone.0197874 and 2021- https://doi.org/10.1017/S0007114520004390) that examine the healthfulness of foods consumed by college students, as well as some economical considerations.
Authors’ response: Thank you for providing these references. We added to the introduction to better reflect the current literature in this area. See lines 42-70
R3. Comment 2. My main concern was not this, but as I continued to read the methods, I see that the authors are using linear regression for a score variable of max 7 points. The distribution of such variable requires careful statistical analysis. Using linear regression in this case is not statistically and epidemiologically sound and the authors should consult with a statistician.
Authors’ response: The regression analysis used average food scores from each student where the average was taken over the number of food items purchased by each student in a semester. We did not use raw scores from each food item as a response. We used multiple linear regression models with heteroscedastic variances inversely proportional to the number of food items purchased in our analysis. We have added these details in the revised version and we made sure to include this information in all table captions.
Round 2
Reviewer 2 Report
The manuscript has been corrected - my comments have been taken into account.
Reviewer 3 Report
I am satisfied with the revised version.